# Joining of $Al_2O_3$ Rods Using Microwaves and Employing Sic Particles as Adhesive

**Natsuko Kimura [1,\*], Takashi Fujii [2], Keiichiro Kashimura [2] and Wataru Nakao [3]**

[1] Graduate school of engineering, Yokohama National University, Yokohama 240-8501, Japan
[2] College of engineering, Chubu University, Kasugai 487-8501, Japan; fujii@isc.chubu.ac.jp (T.F.); kashimura@isc.chubu.ac.jp (K.K.)
[3] Faculty of engineering, Yokohama National University, Yokohama 240-8501, Japan; nakao-wataru-hy@ynu.ac.jp
\* Correspondence: kimura-natsuko-jc@ynu.jp; Tel.: +81-45-339-3692

**Abstract:** The joining of $Al_2O_3$ rods using SiC particles in a microwave field was examined. SiC with high microwave absorption characteristics is coated on the fracture surface of $Al_2O_3$ rods. Then, microwave irradiation is performed using a 2.45 GHz single-mode cavity and the $Al_2O_3$ rods are rapidly joined. Energy dispersive X-ray spectroscopy reveals that the substance generated on the joining surface comprises Al and O. It is believed that the SiC interacts with the microwave to generate microwave plasma and that the plasma melts the $Al_2O_3$ rods. Thus, the matrix melts and the fracture surfaces are joined.

**Keywords:** microwave heating; microwave joining; SiC; $Al_2O_3$

## 1. Introduction

Microwaves, which can heat material rapidly and selectively, have recently become the focus of attention for many researchers. Microwaves can be used to heat the internal components of materials with low thermal conductivity; this property of microwaves has been applied for sintering, relevant chemical reactions, and joining [1–7]. Joining by microwaves in particular has been treated as an important technology by many researchers [8–12]. Ahme et al. reported on the microwave joining of $Al_2O_3$-$ZrO_2$-$SiO_2$ ceramics by employing selective microwave heating. Pulsing microwaves were used to control thermal distribution and an improvement in the joint quality was noticed in their study [8]. Silberglitt et al. recently made advancements in microwave joining by employing SiC. They introduced the direct joining of reaction-bonded SiC (RBSC), together with Si braze results for sintered SiC (SSiC), suggesting the possibility of direct microwave joining of RBSC-SSiC combinations [9]. Fukushima et al. reported the direct joining of $Al_2O_3$-$Al_2O_3$ by microwave heating. Their joining conditions were 3 min and 420 MPa and the average strength of the resultant was equal to the original strength of the material [10]. Aravindan et al. reported that sintered $Al_2O_3$/30%-$ZrO_2$ ceramic composites were joined by hybrid heating using microwave radiation at 2.45 GHz and 700 W, and sodium silicate glass powder as an interlayer. They applied 28 MPa of pressure as well, which improved the joint strength [12].

Microwave joining uses the selective heating of an interface, in which the electrical properties of its two components are the key parameters that control the joining technology. Generally, the absorption of microwave energy by materials is determined by their electrical permittivity and permeability. The microwave power absorbed by materials per unit volume, $P$ [W/m$^3$], is described by $2\pi f \varepsilon_0 \varepsilon_r'' |E^2|$, when the materials are neither metal nor magnetic, where $f$ is the frequency, $\varepsilon_0$ is the permittivity of free space (F/m), $\varepsilon_r''$ is the relative permittivity (imaginary part), and $E$ is the electric field strength. Notably, the absorbed power is largely dependent on $\varepsilon_r''$. When microwave heats a material with $\varepsilon_r''$

selectively, $\varepsilon_r^{''}$ increases with the temperature of the material, and the heated material absorbs more microwave energy than a low temperature material [13–15]. As a result, the joint surface is heated more strongly [16].

The parameter $\varepsilon_r''$ is key to understanding the microwave heating mechanism. For example, it has been reported that $\varepsilon_r''$ of $Al_2O_3$ is lower than that of SiC, and that the microwave power absorbed by the two materials differs [17]. For example, McGill et al. investigated the heating behaviour of materials during microwave irradiation by varying the irradiation power of a 2.45-GHz microwave unit. According to their results, SiC absorbed energy at a temperature of 600 °C after 3 min of being subjected to 2000 W microwave irradiation, while $Al_2O_3$ required the same conditions at an irradiation of 3000 W. Therefore, this indicates that SiC better absorbs microwave energy than $Al_2O_3$. When SiC absorbs microwaves and generates heat, such that it reaches a high temperature, it reacts with the ambient oxygen to produce oxide. Therefore, the possibility of joining under a microwave irradiation can be determined using the relative permittivity (imaginary part).

This study investigated the joining of the rough interface between $Al_2O_3$ rods by employing SiC particles as the microwave absorbing material. We confirmed that SiC is selectively heated by microwave irradiation with the composite material. SiC was applied to the fracture surface of $Al_2O_3$ rods and the rods were adhered to each other. Then, the temperature increase at the fracture surface was investigated. The joining temperature threshold was determined by changing the maximum heating temperature. Finally, in the case of successful joining, we examined the joining ability of SiC and clarified the joining mechanism. The substances generated at the joining interface were analysed and the joining conditions were determined. Hence, the welding mechanism was clarified.

## 2. Materials and Methods

### 2.1. Compound Selection and Fabrication of Specimen

$Al_2O_3$ rods (Otsuka Seiko, SSA-S 99.7%, corner: $10 \times 10 \times 70$ mm$^3$), which has an excellent low $\varepsilon_r'$ and heat resistance, was used as the base material. The adhesive for joining was SiC (Ultrafine, IBIDEN, d = 350 nm), which has a very high $\varepsilon_r''$.

Pre-cracks were introduced to the $Al_2O_3$ rods at 98.07 N (retention time: 15 s) with a Vickers hardness tester. The rods were then broken through three-point bending using a universal testing machine (Autograph AG-X 5 kN, Shimazu Co., Kyoto, Japan).

A SiC slurry, for use as adhesive, was prepared by mixing 1.0 g of SiC and 2.0 mL of isopropanol. The slurry was applied to both fracture surfaces of the $Al_2O_3$ rods. The fracture surfaces were matched to each other and acupressure was applied; then, the slurry was dried. This composite material is referred to as a specimen in this work. Where, the size of $Al_2O_3$ rods was $10 \times 10 \times 70$ mm$^3$ and that of the fracture was about $10 \times 10$ mm$^2$.

### 2.2. Microwave-Irradiation Experiment and Analysis of Joint Behaviour

A microwave heating furnace with a single-mode cavity resonator (Nissin Co. Ltd., Hyogo, Japan) was used [18]. The microwaves oscillating from the magnetron were adjusted by an E-H tuner and introduced to the resonator. The specimen was placed at the electric-field maximum point of the $TE_{103}$ formed between the iris (52 mm) and plunger. The microwave irradiation mode was $E_{max}$. Microwaves as progressive and reflected powers ($P_P$ and $P_R$, respectively) were measured using a power monitor. The interface temperature of fracture-surface was measured using a radiation thermometer (Japan Sensor, FTK9-P300R-30R21, 300 °C $\leq$ Measurement temperature $\leq$ 2000 °C). The spot diameter of the radiation thermometer was 5 mm, and the temperature was measured at the fracture surface. Therefore, the radiation thermometer counted the numbers of photons from the SiC-$Al_2O_3$ interface.

Initially, it was confirmed that the SiC applied to the specimen was selectively heated. Then, experiments with varying holding temperatures were performed and the fracture surface joining was confirmed. After placing the specimen in the microwave heating furnace, $Al_2O_3$ weighing 166.0 g was

placed on top of the specimen and the fracture interface was pressurised. Then, microwave irradiation was performed in conjunction with a dry air flow of 0.6 mL/min. First, in the selective heating experiment, microwave irradiation was performed for 20 min at a constant microwave irradiation power of 600 W. Next, an experiment was conducted to adjust the output temperature by adjusting the microwave output power, so that the heating rate was approximately 50 °C/min. After the experiment was completed, the output power was gradually lowered to prevent a sudden drop in temperature, and adjustment was performed. The output power was adjusted so that the maximum temperature was ±20 °C by checking the radiation thermometer, and the holding time was 1.0 min. Then, the cooling rate was lowered to 5 °C/min. The heating and cooling rates were adjusted to prevent thermal shocking of the ceramics. Microwave heating produces a temperature difference between the surrounding atmosphere and specimen. To maintain a low temperature difference between the specimen and the surrounding atmosphere, the cooling rate was adjusted, where, the temperature distribution of the specimen was measured with a thermo camera (R500Pro, Software: InfReC Analyzer NS9500Pro, Nippon Avionics Co., Ltd, Tokyo, Japan).

The joining strength of the specimen was measured after the joining experiment. Then, the substance involved in the joining was specified. The joining strength was measured via a three-point bending test using a universal testing machine (Autograph AG-X 5 kN, Shimazu Co., Kyoto, Japan) at a crosshead speed of 0.5 mm/min. For substance identification, the composition of the specimen fracture surface was analysed using energy dispersive X-ray spectroscopy (EDX). To observe the ceramic material, which is an insulator in terms of the apparatus specifications, the specimens were coated by sputtering with Pt-Pd alloy or Au. For comparison, the SiC powder applied to the fracture surface before the joining experiment was analysed in the same manner.

## 3. Results and Discussion

### 3.1. Selective Heating via Microwave Irradiation

As explained in the introduction, SiC shows higher microwave absorption than $Al_2O_3$. Considering that the relative permittivity (imaginary part) has frequency dependence [13,14], it can be inferred that SiC with a covalent chemical bond is superior to $Al_2O_3$ with an ionic bond as a microwave absorber. In the following, electrical measurement and their heating behaviour are compared.

From the experimental measurement results of the thermo camera, it is evident that SiC was selectively heated. Figure 1a is a schematic diagram of the specimen used in the experiments. The centre part of the $Al_2O_3$ rods was broken and the SiC adhesive was applied to the fracture surface (see Methods and Appendix Methods). In the fractured side view of Figure 1b, it can be observed that the vicinity of the fracture surface was the brightest. Furthermore, from the observation results of the thermo camera on the fractured side surface in Figure 1c, it was also confirmed that the centre of the fracture surface had the highest temperature. Figure 1d shows the energy and temperature properties of the specimen under microwave irradiation during the experiment, where this temperature was measured by a radiation thermometer, as explained in the methods. The temperature increased in two stages. The blue and red lines correspond to progressive power ($P_P$) and reflected power ($P_R$), respectively. At 175.3 s, the temperature began to rise from 300 °C at a rate of 5.40 °C/s. The temperature increase in the first stage reached 894 °C. Subsequently, the temperature rose gently, reaching 1208.5 °C at 716.1 s. At that time, the $P_R$ reached 17 W. Immediately after the behaviour of this $P_R$ was observed, the temperature began to rise in the second stage at a rate of 4.6 °C/min, to 1373 °C. This type of $P_R$ was often observed by plasma ignition; meanwhile, plasma arcing was probably also present. As seen in Figure 1d, herein, this phenomenon cured at point A, which indicates the sudden increase in $P_R$. Thereafter, the temperature was maintained at almost $1.4 \times 10^3$ °C (1380–1420 °C). The $P_P$ behaviour at 738.9 s stemmed from momentary cessation of the microwave output, before it instantly restarted. After the microwave irradiation was stopped, the temperature decreased suddenly, where the thermometer used for this measurement cannot measure 300 °C or less. Figure 1e shows the

temperature distribution of a specimen heated by microwaves measured from the side. As can be seen from the counter images, the centre temperature is high in each image, indicating that the temperature conducts from the fracture surface to $Al_2O_3$ rods. Therefore, the SiC applied to the fracture surface is selectively heated.

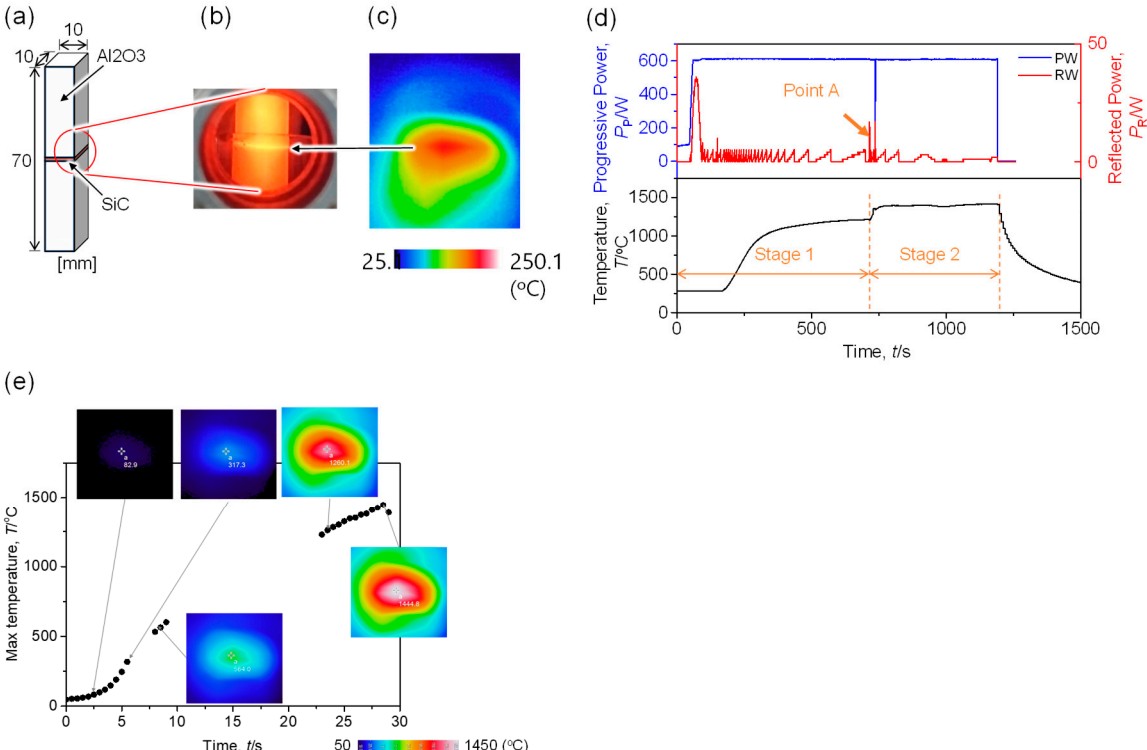

**Figure 1.** A result of an experiment to test selective heating behaviour. (**a**) Schematic diagram of specimen shape. SiC was applied by breaking the centre position of the $Al_2O_3$ rods. (**b**) Selective heating state at a constant microwave output power of 600 W. The fracture surface position is brightest, indicating high temperature radiating from the SiC. (**c**) Observation of fracture surface temperature during heating using thermo camera. The temperature rise was confirmed from the fracture centre. (**d**) Energy characteristics (upper figure) and heating behaviour (lower figure) under microwave irradiation. The blue and red lines indicate progressive ($P_P$) and reflected ($P_R$) power, respectively. (**e**) Thermal distribution of the joining interface measured via thermo camera.

Although the temperature gradient decreases as temperature increases, it is surmised that the microwave absorption of the SiC increases. As shown in the equation $P = 2\pi f \varepsilon_0 \varepsilon_r'' |E^2|$, microwave absorption is a function of the relative permittivities (imaginary part) of the materials. As the temperature rises in SiC, the imaginary part decreases (or stagnates) once, but rises again above 1000 °C [13,14]. Therefore, at 1000 °C or higher, the microwave selectively heats SiC more significantly. In addition, the sudden increase in reflected powers observed at point A in the figure indicates that phenomena other than the increase in the imaginary part are occurring. Considering that the reflected power and temperature discontinuity were observed at point A, it is suspected that micro plasma has appeared at the joining interface.

### 3.2. Identification of Substances Affecting Joining

$Al_2O_3$ rods were melted by selective heating of SiC at the fracture surface. Figure 2a shows the specimen shape after microwave irradiation. A fracture surface can be confirmed at the centre of the $Al_2O_3$ rods, and SiC is applied to this fracture surface. Figure 2b,c shows the photographs of the fracture surface shape after and before the microwave irradiation, respectively. Before the microwave irradiation, SiC powder, which are black, was pasted to the fracture surface. Microwave irradiation

turned this to white as shown in Figure 2b and created a crack in the x-axis direction. As shown in this figure, melting materials were observed beside cracks, which has some height. Figure 2d shows the results of element mapping by SEM-EDX. Where, the surface analysis results are a combination of Al, Si and O. A lot of Al was detected in the centre of the analysis range (shown by yellow colour), and a lot of Si was detected in the blue part. These results and Figure 2e,f indicate that the melted materials were $Al_2O_3$ that oozed out from the fracture surface. Furthermore, some Si elements probably existed at the interface as mullite ($Al_6O_{13}Si_2$) phase considering that there are Si elements.

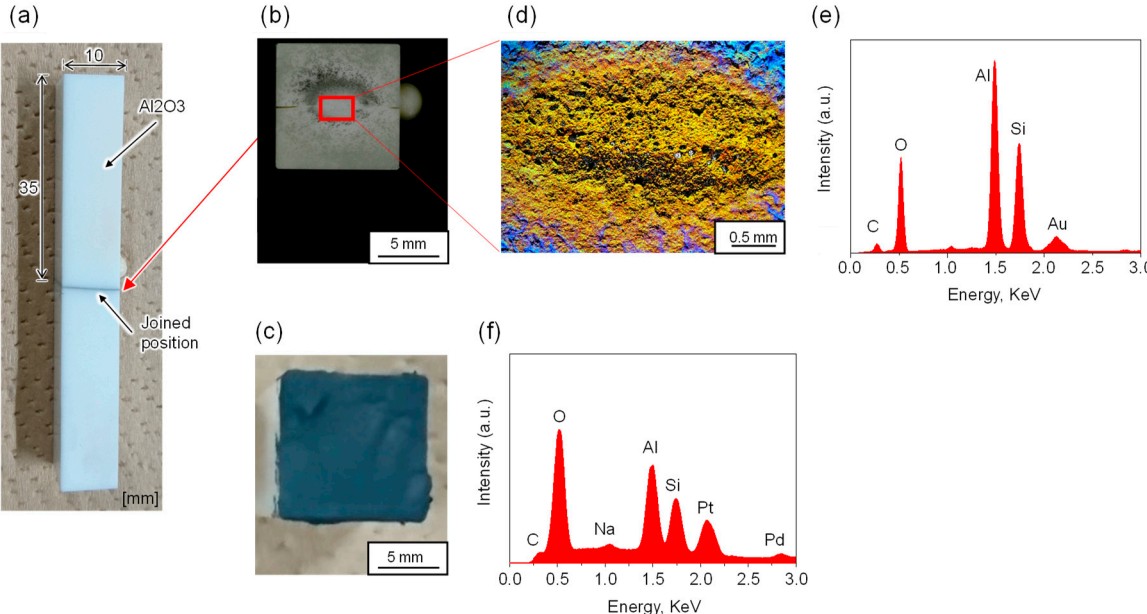

**Figure 2.** Elucidation of substances affecting the joining. (**a**) Specimen shape after joining. (**b**) Fracture surface observation after the test; the red square is the energy dispersive X-ray spectroscopy (EDX) analysis range. (**c**) Photograph of fracture surface coated with SiC before the experiment. (**d**) Results of EDX analysis overlay at Al and Si, O. Yellow area represents Al elements and blue, Si. (**e**) EDX elemental analysis results. Au, Pt, and Pd were used for sputtering. (**f**) EDX elemental analysis results for SiC powder.

*3.3. Elucidation of Joining Mechanism Due to Microwave Irradiation*

Point A makes a significant contribution to the success or failure of the joining. Figure 3a shows the point A appearance range and the joining temperature threshold, which was determined from the joining experiment results. As shown in the figure, point A was not observed at temperatures lower than 1200 °C. Figure 3b shows the energy and temperature properties of the microwave corresponding to successful joining. The black line shows the temperature measured by the radiation thermometer, the blue line shows the $P_P$, and the red line shows the $P_R$. The behaviour of point A was also determined in this experiment. The temperature began to exceed 280 °C at 532 s. It rose at 7.2 °C/s after approximately 3 s, when point A occurred, and reached 1310 °C. Thereafter, the temperature was held for 1 min. Point A momentarily peaked in a value of 95 W at 1282 °C. The output power at that time was 844 W. In the measurement where point A as shown by this curve exists, the $Al_2O_3$ rods were joined, and this tendency has been observed for all measurements. Figure 3c shows the results corresponding to joining failure. For those specimens, all the SiC applied to the fracture surface reacted to become white-coloured, but there was no cavity in the centre. The temperature increased to 280 °C at 473 s and reached 1300 °C at 1900 s. It was then maintained at 1300 ± 5 °C and then decreased after 1 min. In this experiment, the maximum output was 752 W, although point A was not observed. The rising edge of the $P_R$ seen at 1960 s was not Point A, because it was generated by the reduced $P_P$ output. Therefore, the rise of this reflected power is not point A because it did not show a rapid temperature rise. Figure 3d

shows the holding temperature and holding time for each joining experiment, the output power and temperature at which point A was generated, and the joining strength. No. 1–3 is SiC (Ultrafine, IBIDEN), No. 4–No. 13 is the result of SiC (include $Na_2O_3$) used in Appendix A. Figure 3b,c is Test No. 3 and No. 6, respectively. As described above, even when the temperature was the same, when the behaviour of point A did not occur, no joining was made. From these results, we conclude that the typical behaviour at point A is necessary for the joining to occur. We considered the possible causes of the phenomenon by which reflected powers and temperature increase rapidly at point A. When point A appeared, there was a sudden rise in temperature; therefore, and because of the heat transfer rate relationship as shown in Figure 1d, it is conceivable that the centre of the specimen had a higher temperature than the surface. The selective heating of the joining interface was observed by plasma or thermal runaway. With the assumption that the plasma was generated at the joining interface, the plasma gas is estimated to be $CO(g)$. According to equilibrium thermodynamics, $SiO(g)$ and $CO(g)$ exist on the joining interface, where the reaction constants are: $SiC + 3/2O_2 \rightarrow SiO_2 + CO$ : 28.85 and $SiC + O_2 \rightarrow SiO + CO$ : 13.56 in there log $K_f$ [19]. Further, the $SiO(g)$ was generated to the same extent. Moreover, although $SiO(g)$ was generated, its amount was less than that of the generated $CO(g)$, according to the reaction constant. Therefore, it can be concluded that $CO(g)$ plasma is generated at the joining interface and heated. With the consideration that mullite is produced at 1400 °C or higher [20], it can be estimated that the joining interface was heated to 1400 °C or higher. This hypothesis can successfully explain the increase in reflected powers at point A. Another hypothesis is based on the SiC thermal runaway. It has been reported that when a mixed material is heated by microwaves, only the material that has good microwave absorption is selectively heated. This is caused by an increase in the imaginary part of the relative permittivity because of the temperature rise [16]. Since SiC absorbs microwaves at high temperatures [13], the system in this study meets the required conditions.

When claiming abnormal heating of the joining interface by selective heating, it is difficult to explain the increase in reflected powers at point A. As shown in Figure 3e, point A is thought to have occurred owing to the fact that substances that absorb microwaves change because of oxidation and the phase of the microwaves is shifted. Initially, when the specimen was irradiated, the microwave power was absorbed via SiC, which has a higher imaginary part of the relative permittivity than $Al_2O_3$, and subsequently converted into heat. This, in turn, increased the temperature of SiC, which caused the gradual oxidation of SiC in the air, resulting in $SiO_2$. Reportedly, the relative permittivity (imaginary part) of $SiO_2$ is lower than that of SiC [21], which causes the decrease in the absorption of microwaves at the joining interface of the transition of SiC to $SiO_2$. SiC, as a semiconductor, releases thermionic electrons at high temperatures [22]. Therefore, these results indicate that the phenomenon observed at point A is due to the plasma arcing, which was triggered by the electrons emitted by SiC. It is conceivable that $CO(g)$ instead of $SiO_2$ reacted with the electrons to generate plasma. When plasma is irradiated by microwaves, the microwave wavelength changes in the plasma [23,24]. Furthermore, the microwaves are reflected when the plasma becomes dense. Therefore, it can be inferred that point A developed because of the phase shift associated with the plasma. From the thermodynamic perspective, it can be predicted that there was high CO partial pressure on the joining interface. As there were many kinds of gases and powders on the joining interface, there is a high possibility that the plasma was excited by the microwaves. It is thought that the temperature rises in the second stage became even higher because of reaction heat from the plasma formation.

By using microwave plasma in this manner, it is possible to locally raise the temperature to melt the $Al_2O_3$ rods. The plasma with increasing temperature contributes greatly to the joining.

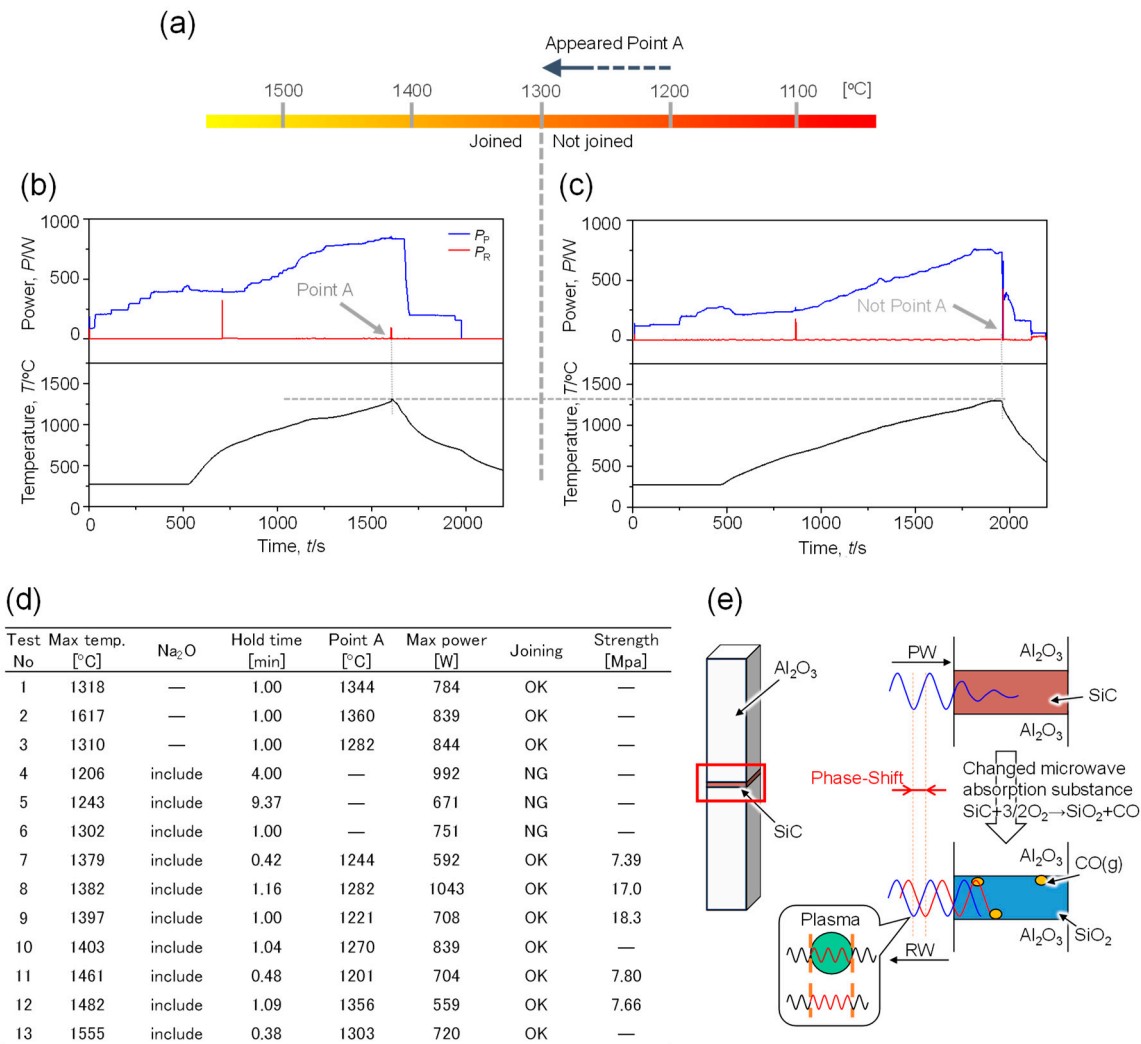

**Figure 3.** Joining experiment results. (**a**) Temperature bar showing threshold range for $Al_2O_3$ rods joining and range in which point A appeared. (**b**) Experimental results for successful joining: point A appeared. (**c**) Experimental results for joining failure: point A did not appear. (**d**) List of joining experiment results: point A appeared upon joining. Note that the No. 3 and No. 1 results are shown in Figure 3b,c, respectively. (**e**) Behaviour of point A when it appeared.

## 4. Conclusions

To develop a novel joining method using the microwave irradiation, the joining of $Al_2O_3$ rods was examined. SiC with high microwave absorption is coated on the rough fracture of sintered $Al_2O_3$ rods and then, microwave irradiation is performed using a 2.45 GHz single-mode cavity. Thermography results suggest that SiC particles were selectively heated by microwave irradiation. The imaginary part of the electrical permittivity increased with temperature, so the heating rate of SiC increased exponentially. Considering that the reflected power at $1.3 \times 10^3$ °C has a great influence on the joining result, it is inferred that plasma generation is necessary for joining.

**Author Contributions:** Conceptualization, N.K.; K.K.; data curation, T.F.; formal analysis, N.K.; T.F.; funding acquisition, N.K.; K.K.; W.N.; investigation, N.K.; project administration, W.N.; resources, K.K.; W.N.

**Funding:** This research was funded by Japan Society for the Promotion of Science grant number 17J11441.

**Acknowledgments:** We gratefully appreciate the financial support of Tateho Chemical Industries Co., Ltd. that made it possible to complete this study.

**Conflicts of Interest:** There are no other relationships or activities that could appear to have influenced the submitted work.

## Appendix A. Joining of Al$_2$O$_3$ Rods Using Microwaves and Employing Sic Particles as Adhesive

*A.1. Appendix Methods*

*A.1.1. Relative Permittivity Measurement of Al$_2$O$_3$ Rods and SiC as the Adhesive, and Heat Generation Properties because of Microwave Irradiation*

Al$_2$O$_3$ rods (Otsuka Seiko, SSA-S 99.7%, corner: $10 \times 1 \times 70$ mm$^3$), which has excellent low $\varepsilon_r''$ and heat resistance, was used as the base material. SiC was used as adhesive (acicular powder: 30–200 μm) [15] because it has good microwave heat generation characteristics. The SiC powder was micronised using a bead mill (LABSTAR min, Ashizawa Fintech Ltd., Chiba, Japan, i).

The SiC was further micronised so that the SiC diffused into the rods. Pulverisation was performed using a bead mill with alumina beads. Then, 5.06 g of SiC, 100 mL of isopropanol, and 50.84 g of alumina balls were placed in a bottle of soda glass and pre-ground via ball milling. Note that trace amounts of NaO$_2$ were contained in the powder because of its preparation method. Next, the slurry resulting from the ball milling was pulverised by a bead mill. As the SiC exhibited sufficiently good microwave absorption characteristics, it was mixed with Al$_2$O$_3$ supplied from beads, such that the SiC content was 4.0 wt.%. The average particle size after bead milling was measured using a zeta potential meter (Nano-ZS, Malvern Panalytical Ltd., Malvern, Worcestershire, United Kingdom) and was found to be 295 nm. Finally, the SiC slurry was dried and the powder was collected.

For the relative permittivity measurement, a perturbation method [25] based on a cylindrical cavity (TM$_{010}$ mode) was used. The powder sample was inserted into a quartz holder of φ2 × 25 mm. The sample weights were 0.070 and 0.1348 g for the SiC and Al$_2$O$_3$, respectively. Only the quartz holder was inserted into the cylindrical cavity, and the resonance frequency and $Q$ value were measured using a network analyser. The powder was then inserted into the cylindrical cavity and measurements were performed in the same manner. From the changes in the two resonance frequencies, the real and imaginary parts of the relative permittivity of the SiC and Al$_2$O$_3$ powder were clarified. The perturbation coefficient used here was 1.855.

A single-mode cavity resonator (Nissin Inc., Hyogo, Japan) [18] was used as the heating furnace. The microwaves oscillating from the magnetron were adjusted by an E-H tuner and introduced to the resonator. The Al$_2$O$_3$ rod and SiC powder were placed in the electric-field with maximum point of the TE$_{103}$ formed between the iris (52 mm) and plunger. The electric-field absorption characteristics of the sample were then evaluated. The conditions were as follows: Microwave irradiation power: 220 W; microwave irradiation duration: 20 min; dry air flow: 0.6 mL/min constant. SiC powder (0.5 g) was used to fill a φ12 quartz tube and placed in a furnace. The temperature rise was measured by irradiating the radiation thermometer from the observation window. Note that the radiation thermometer could not measure temperatures lower than 300 °C, because of the device specification. The Al$_2$O$_3$ rods was placed in the furnace, and a fibre thermometer was used for the measurement. As a fibre thermometer melts at 300 °C or higher, the microwaves were stopped when the temperature reached 250 °C.

*A.1.2. Effect of Na$_2$O$_3$ Content on Joining Behaviour*

Joining behaviour by impurities was confirmed using SiC containing Na$_2$O$_3$ by micronisation. SiC was applied to the fracture surface of the Al$_2$O$_3$ rods and dried in a state where the fracture surface was matched to prepare a specimen. The specimen was placed in a furnace, and Al$_2$O$_3$ weighing 166.0 g was placed on top of the specimen to pressurise the fracture surface. Then, microwave irradiation was performed in conjunction with a dry air flow at 0.6 mL/min and held for 1 min after point A appeared. The joining behaviour of the fracture surface and the temperature rise curve were compared with SiC (ultrafine, IBIDEN Co., Ltd., Gifu, Japan) not containing Na$_2$O$_3$.

## A.2. Appendix Results and Discussion

### A.2.1. Relative Permittivity Measurement of Al$_2$O$_3$ Rods and SiC as the Adhesive, and Heat Generation Properties because of Microwave Irradiation

We found that SiC works well as an adhesive. Figure A1 shows the heat generation properties when the sample was irradiated by microwaves at a constant output power. The conditions other than the output at the time of irradiation are shown on the upper left of Figure 2. Note that the radiation thermometer used for SiC measurement was incapable of measuring temperatures lower than 300 °C, because of its specifications. Therefore, the temperature result indicates a constant value of 300 °C until this temperature was exceeded. The dashed lines indicate the time at which the microwave irradiation was stopped, and the triangular symbols show the temperature rise slopes. The $\varepsilon_r'$ and $\varepsilon_r''$ values in room temperature were 2.44 and 0.144 for SiC, respectively, and 2.75 and 0.0230 for Al$_2$O$_3$, respectively. The SiC heat generation was divided into three stages: temperature increase, steady state, and cooling. During the heating process, the temperature began to exceed 300 °C at 42.3 s. A temperature of 520.8 °C was achieved at 62.8 s, after increases of 16.1 °C/s during the temperature increase stage. The SiC heating rate was higher than that of Al$_2$O$_3$, which was 0.189 °C/s. The SiC temperature reached 836.1 °C at 600 s, after gradual increases. In the steady state, the temperature was constantly maintained at 839.5 ± 4 °C. This was due to the heat balance equilibrium caused by the heat generation because of the microwave absorption, and the heat release to the surrounding atmosphere. In the cooled state after microwave termination, the temperature decreased at a rate of 3.61 °C/s. The heat generation was stopped by terminating the microwave irradiation; this is attributable to the fact that the heat removal expenditure was larger. The Al$_2$O$_3$ increased to 500 °C at 0.189 °C/s, and then exponentially increased until 745.8 s, reaching 251.7 °C. The microwave irradiation was then stopped, and the temperature decreased rapidly at a cooling rate of 0.223 °C/min.

The cause of the high SiC heat generation rate is the higher $\varepsilon_r''$ of SiC compared to Al$_2$O$_3$. Therefore, the SiC heat generation calculated from the heat generation equation, *P*, exceeded that of Al$_2$O$_3$. This result implies that the SiC absorption exceeded that of Al$_2$O$_3$. Jia et al. have reported that SiC oxidation begins at approximately 800 °C and gradually produces SiO$_2$ [26]. In the steady state, although the temperature was held at 800 °C or more, at which SiC oxidation was predicted, chemical reaction other than adhesives was not observed and the material was stable. It is thought that the heat of the chemical reaction was low because the SiC content was small, at 4 wt.%. Furthermore, the rapid temperature decrease in the cooling state revealed that the temperature of the atmosphere surrounding the sample was lower than that of the sample. Hence, sufficient microwave absorption for reaction and heat generation was confirmed, even for SiC content as low as 4 wt.%.

### A.2.2. Effect of Na$_2$O$_3$ Content on Joining Behaviour

Figure A2 shows the results of the joining experiment, (a) SiC simple substance (Na$_2$O$_3$ not include) (b) SiC (include Na$_2$O$_3$). Red line is $P_R$, blue line is $P_P$, black line is temperature rising curve. The arrow shown at the bottom of the temperature rise curve shows the behaviour of the temperature rise curve. Point A occurred in both cases, and the temperature was raised, and the behaviour of rapid temperature rise was observed. However, it showed different behaviours when holding for 1 min after rapid temperature rise. In Figure A2a, the temperature was lowered rapidly and the SiC remained unoxidized on the side fracture surface. On the other hand, in Figure A2b, the temperature decrease after the rapid temperature rise was suppressed, and all fractured SiC was oxidized.

This is because the microwave energy was absorbed at a position different from the irradiation position of the radiation thermometer which was measuring the temperature of the side surface of the fracture surface as the factor of lowering the temperature during holding with SiC alone. By containing Na$_2$O$_3$, it is considered that plasma generated by heating of SiC induces microwave energy in the fracture surface because Na$_2$O$_3$ is next converted into plasma. It is known that SiC as a semiconductor releases thermionic electrons at high temperatures [22]. Na as the alkali metal has a first ionization energy of 5.13

eV [27], which is lower than the ionization energy of 14.1 eV [27] of CO. From this, it is understood that Na (g) reacts with electrons to generate plasma easily. For this reason, plasma generated by SiC triggers Na as a plasma, so it is considered that the fracture surface was maintained at a high temperature state.

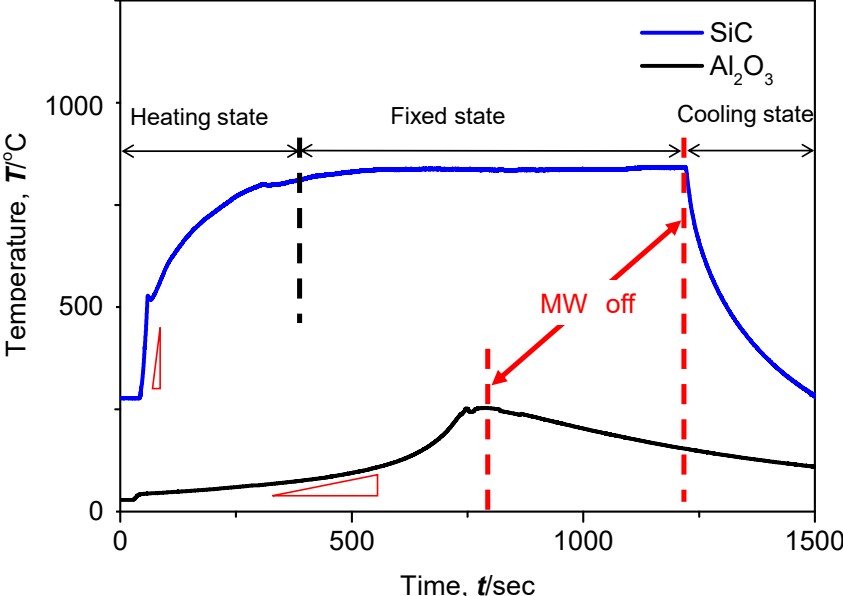

**Figure A1.** Heat transfer characteristics of SiC and $Al_2O_3$ for microwave irradiation with constant output power. The dry air flow was 5 mL/min and the microwave was constant at 220 W with $E_{max}$. The blue and black lines indicate the results for the SiC powder and $Al_2O_3$ rods, respectively. The red dotted line indicates the times at which the microwave irradiation was stopped. Each red triangular symbol indicates the heating speed because of heat generation. A steeper slope corresponds to a higher generated heating speed.

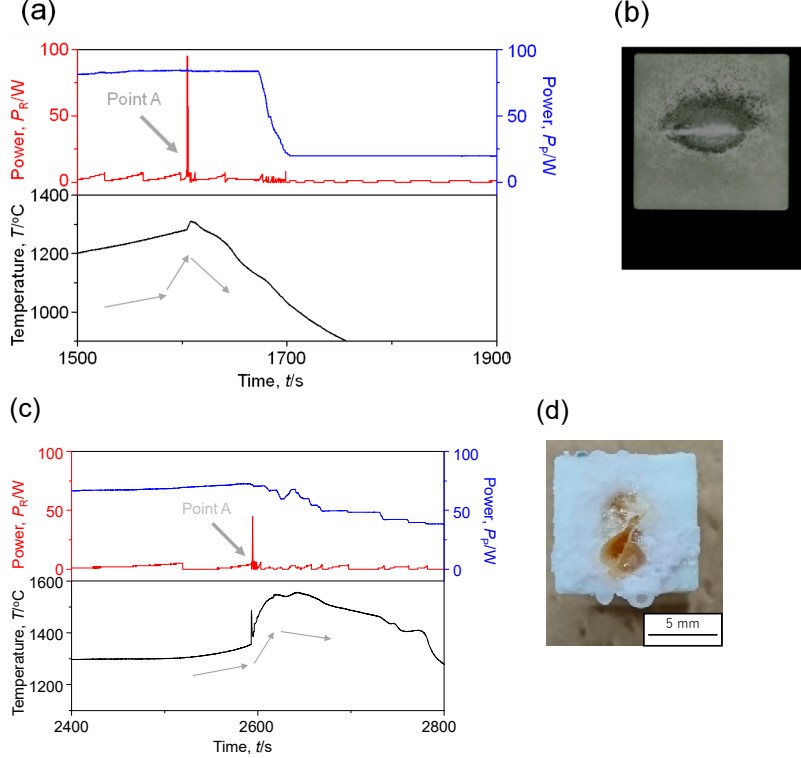

**Figure A2.** Influence of $Na_2O$ content on SiC single body and SiC joining behaviour. The red line in the figure is $P_R$, the blue line is $P_P$, and the black line is the temperature rising curve. (**a**) Joining

experiment result of SiC simple substance. (**b**) Photograph of fracture surface after the experiment of (**a**). (**c**) Joining experimental results when Na2O3 was contained in SiC. (**d**) Photograph of fracture surface after the experiment of (**c**). Since the behaviour of point A was observed in (**a**), it was proved that the generation of plasma was due to SiC. In (**b**), since the temperature decrease after the rapid temperature rise was more gradual than (**a**), and all of the fractured SiC was oxidized, it was possible to induce plasma to the fractured surface by containing $Na_2O_3$.

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
