# Peer review of "Joining of Al2O3 Rods Using Microwaves and Employing Sic Particles as Adhesive"

_processes, doi:10.3390/pr7100750_

Round 1

Reviewer 1 Report

Kimura et al report on using SiC particles under microwave to “self-heal” a fracture in alumina substrates. However, what they describe as “self-healing” is not exactly what one might think. The SiC has to be deposited externally after the fracture is formed in alumina. This can’t be called self-healing as it requires another material and an external step to “heal”.

·       On page 2, the mention “... Thus, self-healing with the same functionality as the conventional self-healing mechanism is possible. In addition, a high temperature atmosphere can be created uniquely, with no dependence on the environment.” To me, this approach is not a conventional self-healing process. First, the fracture must be diagnosed somehow, and then deposition of another material (SiC) will follow, and subsequently the mix will be exposed to microwave. In conventional self- healing, no such intervention is required. Any exposure to external stimuli will autonomously heal the substance.

·       What is the “arrival temperature”? Page 2, line 94.

·       In their experiment on page 3, the authors should explain how they confirmed that SiC was selectively heated? Is it theoretical or experimental? Selective heating experiment is not clear either.

·       How large was the fracture?

·       More explanation is required for measurement of the bond strength.

·       Was the thermography done during microwave exposure? If so, the authors should provide some explanation on the experimental set up.

·       Figure 2c, they mention measurement error, what do they mean by that? Figure 2c should be colour coded.

·       Overall, the description of Figure 2 is very confusing.

·       Si and C are still present in figure 2d, despite the authors claim in the abstract, that they are gone after microwave. Is it expected for the SiC to disappear? To where?

·       They refer to point A. What is point A. The description is not clear.

·       The mechanism of different state microwave heating is not discussed clearly. Was a systematic experiment?

·       What is the advantage of this technique to welding (if another material (SiC slurry) has to be added.

·       While I believe, this work could be interesting, the way the paper is written makes it very difficult to understand the discussion.  It lacks clarity in defining many terms. It doesn’t read well. The authors need to improve the paper both in terms of the writing style if I may say, and the clear description of the terms they use to make it comprehensive for readers, and publishable.

Reviewer 2 Report

This paper describes microwave joining experiments of Al2O3 beams with a SiC interlayer. The authors observe a sudden increase in temperature at some point of the microwave heating process, combined with a short peak in the reflected wave. In the EDX analysis of the joining interface, no (or trace amount of) Si and C are found. The authors conclude that SIC is converted to gaseous products upon oxidation, and that some of these gases get ionized by the microwave field and form a plasma which melts the Al2O3 and provides a good joint.

To start with, I would like to note that there has been a significant body of work on microwave joining of ceramics published since 1990s. This paper could be improved by briefly reviewing this background in the introduction. The paragraph of the introduction describing the microwave heating (lines 64 - 78) should definitely be improved. In the current form, this descrption is superficial, fragmentary, lacks the relevant references and even contains errors. For example, the formula in line 70 is incorrect (probably two systems of units are mixed in it) and has a wrong description (Q is not an energy but rather a power).

Regarding the abrupt temperature increase phenomenon, it is rather frequently observed in microwave processing. Most often it is associated with a thermal instability originating from the rising dependence of epsilon'' on T. In this connection, it should be mentioned that the authors are apparently not aware of the existence of such a dependence. For example, in line 84 they cite the values of epsilon'' of some materials and do not mention the temperature at which they were measured. In a similar manner, they use temperature-independent dielectric properties of materials in the modeling of microwave heating (lines 173-196). In reality, the values of epsilon'' may change by orders of magnitude during the heating process, which affects not only the absorbed power, but also the distribution of the electromagnetic field in the material. The proposed plasma formation is not substantiated in the paper by any experimental observations or measurements. Because of that, the proposed mechanism does not look convincing.

Finally, the paper has serious flaws both in the English language and in the preparation of the manuscript. For example, even in the first sections of the paper I could find a number of errors:

- line 88: does "uniquely" mean "locally"?

- line 100: epsilon' should be replaced with epsilon''

- line 100: having looked at Fig. 2b, I would guess that the real dimensions of the sample are 10 x 10 x 70 mm3

- line 128: could not understand what does "+/- 20 deg C" mean here

- line 151: change "was" to "had"

- line 154: Fig. 1 - change (c) to (d)

- line 159: electromagnetic waves are generally reflected, to a certain extent, from the surface of any material, not limited to the materials having free electrons

- line 177: Fig. 1 - change (d) to (e)

- line 178: Fig. 1 - change (e) to (f)

- line 180: Fig. 1 - change (d) to (e), (f) to (g)

- line 228: Fig. 2 - change (c) to (b)...

My recommendation is that the authors should rewrite the paper in order to provide a more thorough review of the microwave heating fundamentals and the previos work on microwave joining, improve the English language and the quality of presentation, and reconsider the mechanism of the described phenomenon.

Reviewer 3 Report

This is a good and interesting paper and might be published as it stands.

Author Response

We deeply appreciate the efforts of the reviewers for reviewing our manuscript. They have improved the presentation of our paper. We agree with the comments and have revised the manuscript accordingly. All major revisions suggested by the reviewers are shown in red font in the revised manuscript.

Reviewer 4 Report

The manuscript is ready to publish.

Author Response

(The authors gave the same response as above.)

Round 2

Reviewer 1 Report

The authors made an effort to improve the quality of the paper. However, my concerns for this paper have not been fully addressed. I do not agree with the term “self-healing” for the reasons I previously outlined. To me, this approach is not a conventional self-healing process. First, the fracture must be diagnosed somehow, and then deposition of another material (SiC) will follow, and subsequently the mix will be exposed to microwave. In conventional self- healing, no such intervention is required. Any exposure to external stimuli will autonomously heal the substance.

In page 3, line 128, why the temperature rises at point A? What is the mechanism. It’s not clear if the two stages temperature is the cause or effect.

In Figure 1d, how is the temperature measured and plotted? Is this from the thermo camera or from the microwave IR sensor? If it’s the former, which part of the sample has been measured? Is that SiC? How the temperature on a specific point on the sample was recorded? If it’s the latter, microwave temperature sensor does not necessarily reflect the sample’s temperature. Sample’s temperature should be measured independently. In page 4 line 138, they mention “Where the thermometer used for this measurement cannot measure 300 ° C or less.” Is that correct? They should provide the spec of the thermometer.

Numerical calculation in figure 1e show the result along x axis., in other words, in the plane that SiC is located, and it goes from 1160℃ to 1320℃. In figure 1f, along the y axis, the temperature range is exactly the same 1160-1320℃. Is this a coincidence? More importantly, it’s not clear the y-axis range in Fig 1f include Aluminium oxide, or it is exclusively for SiC? Either way it doesn’t match the experimental result in figure 1c, showing the temperature range between 25-250℃. The reason of this discrepancy is not explained clearly.

The new lines added in page 5 line 178- 182 are a bit confusing. It’s written “ The rapid temperature rise at Point A in Fig. 1(d) was not revealed by the numerical calculation because it did not consider the temperature dependence of electrical permittivity”. Isn’t figure 1d showing the experimental result?? The language is very confusing.

The newly added lines 206- 207 are not clear. Does it mean the they mix the material where “point A” appeared in fig 1d? The sentence needs to be corrected. Also, describing “point A” properly and referring to a proper scientific term is more accurate than calling it “point A” through the paper.

Although the authors made an effort to improve the paper, it requires further work in particular in writing style to make it more understandable.

The authors made an effort to improve the quality of the paper. However, my concerns for this paper have not been fully addressed. I do not agree with the term “self-healing” for the reasons I previously outlined. To me, this approach is not a conventional self-healing process. First, the fracture must be diagnosed somehow, and then deposition of another material (SiC) will follow, and subsequently the mix will be exposed to microwave. In conventional self- healing, no such intervention is required. Any exposure to external stimuli will autonomously heal the substance.

In page 3, line 128, why the temperature rises at point A? What is the mechanism. It’s not clear if the two stages temperature is the cause or effect.

In Figure 1d, how is the temperature measured and plotted? Is this from the thermo camera or from the microwave IR sensor? If it’s the former, which part of the sample has been measured? Is that SiC? How the temperature on a specific point on the sample was recorded? If it’s the latter, microwave temperature sensor does not necessarily reflect the sample’s temperature. Sample’s temperature should be measured independently. In page 4 line 138, they mention “Where the thermometer used for this measurement cannot measure 300 ° C or less.” Is that correct? They should provide the spec of the thermometer.

Numerical calculation in figure 1e show the result along x axis., in other words, in the plane that SiC is located, and it goes from 1160℃ to 1320℃. In figure 1f, along the y axis, the temperature range is exactly the same 1160-1320℃. Is this a coincidence? More importantly, it’s not clear the y-axis range in Fig 1f include Aluminium oxide, or it is exclusively for SiC? Either way it doesn’t match the experimental result in figure 1c, showing the temperature range between 25-250℃. The reason of this discrepancy is not explained clearly.

The new lines added in page 5 line 178- 182 are a bit confusing. It’s written “ The rapid temperature rise at Point A in Fig. 1(d) was not revealed by the numerical calculation because it did not consider the temperature dependence of electrical permittivity”. Isn’t figure 1d showing the experimental result?? The language is very confusing.

The newly added lines 206- 207 are not clear. Does it mean the they mix the material where “point A” appeared in fig 1d? The sentence needs to be corrected. Also, describing “point A” properly and referring to a proper scientific term is more accurate than calling it “point A” through the paper.

Although the authors made an effort to improve the paper, it requires further work in particular in writing style to make it more understandable.

Reviewer 2 Report

I recognize that the authors have attempted to address the criticism that was contained in my first referee report; however, unfortunately, I am not satisfied with the changes that have been introduced. In particular, the parts of the paper that describe the physics of the microwave interaction with the materials remain incorrect, as explained below.

Once again, "the absorption energy" mentioned in lines 32-33 is actually the absorbed power and not the energy.

The formula in line 33 (also repeated in line 181) is correct only if epsilon'' is meant as a dimensional quantity, i.e. it is a product of epsilon_0 (the electric constant, equal to 8.85*10^-12 F/m) and relative epsilon''. However, everywhere else in the paper epsilon'' is dimensionless (for example, in line 44: "SiC has an epsilon'' of 27.99"). To correct this error, the electric constant epsilon_0 should be inserted into the formula as an additional multiplier.

Line 35: the electric field amplitude E in the material is actually NOT constant. As implied by Maxwell's equations, the distribution of the field depends on the distribution of the dielectric permittivity of the material which changes as the temperature increases.

Line 39: I cannot see why lambda_0 and epsilon_0 are mentioned here.

Line 40: The denominator contains epsilon'' and not epsilon'.

I still have concerns about the relevance of the electromagnetic simulation described in the paper. Once again, the simulation that does not account for the temperature dependence of epsilon'' cannot provide meaningful results when simulating processes with rapid changes of temperature. In addition, I cannot understand how the authors choose the values of the simulation parameters. For example, it is mentioned in line 44 that SiC has an epsilon'' of 27.99; however, for the simulation epsilon'' of this material is taken equal to 0.144 (Figure caption for Fig. A1). Similar discrepancies exist for other parameters.

My suggestion at this point is to remove all the simulations of the microwave heating from this paper because they are apparently not relevant. Also, I find it doubtful that the suggested mechanism of the observed phenomenon really takes place. Nevertheless, the paper could still be interesting from the experimental point of view. 

Round 3

Reviewer 2 Report

The authors have excluded the electromagnetic simulation from the paper in accordance with my suggestion in the previous round of review. Also, they attempted to rewrite the part of introduction that describes the electromagnetic background behind the microwave heating. This part, however, still has to be improved. My suggestions for doing so are listed below:

line 16: Microwaves ... haVE become...

line 33: delete the second "are"

lines 33-34: replace the sentence "The ceramics heated by microwaves absorbed their energy via electrical or magnetic properties" with something like "The absorption of microwave energy in materials is governed by their dielectric and/or magnetic properties."

lines 34-35: replace the fragment "The absorption power of materials in microwaves" with "The microwave power absorbed in materials"

lines 38-39: replace the fragment "While " is a product of the dielectric constant ’0 and ??" is the relative permittivity (imaginary part) [13], absorption" with "The absorbed"

lines 40-41: delete the fragment "a change in the dielectric constant changes the electric field strength inside the substance," because it sounds misleading.

lines 43-45: delete the fragment "In addition, the microwave penetration depth D is determined by ?=?√??′ ???"⁄, where c is the speed of light and ’r is the relative dielectric constant [18]. Therefore, the microwave penetration depth is also an important parameter for microwave joining", because it does not help to understand the paper.

lines 48-49: The authors still have not removed (nor explained) the discrepancy between the epsilon'' values listed here, in Table 1, and in the caption to Figure 2. (They provided some explanation in the response to the reviewer but not in the manuscript). I would suggest deleting all quantitative values of epsilon'' from the manuscript, because they are material-specific and temperature-dependent, and generally it does not make much sense to mention them with such an accuracy (like 27.99). Instead, it would be sufficient to mention that alumina has generally a much lower microwave absorption than SiC, and provide a reference.

line 133: The denomination "point A" chosen by the authors is difficult to understand in its current form. I guess that they have chosen this letter because usually the sudden increase in the reflected power is a signature of arcing in the system. Perhaps it would be good to explain somewhere in the paper that "A" stands for "arcing".

Finally, the English language should be once again checked thoroughly throughout the entire paper.
